# Nanodrug rescues liver fibrosis via synergistic therapy with $H_2O_2$ depletion and Saikosaponin b1 sustained release

Mengyun Peng[1,4], Meiyu Shao[1,4], Hongyan Dong[1], Xin Han[1], Min Hao[1], Qiao Yang[1], Qiang Lyu[1], Dongxin Tang[2], Zhe Shen[3], Kuilong Wang[1], Haodan Kuang[1] & Gang Cao [1✉]

Hypoxia and hydrogen peroxide ($H_2O_2$) accumulation form the profibrogenic liver environment, which involves fibrogenesis and chronic stimulation of hepatic stellate cells (HSCs). Catalase (CAT) is the major antioxidant enzyme that catalyzes $H_2O_2$ into oxygen and water, which loses its activity in different liver diseases, especially in liver fibrosis. Clinical specimens of cirrhosis patients and liver fibrotic mice are collected in this work, and results show that CAT decrease is closely correlated with hypoxia-induced transforminmg growth factor β1 (TGF-β1). A multifunctional nanosystem combining CAT-like $MnO_2$ and anti-fibrosis Saikosaponin b1 (Ssb1) is subsequently constructed for antifibrotic therapy. $MnO_2$ catalyzes the accumulated $H_2O_2$ into oxygen, thereby ameliorating the hypoxic and oxidative stress to prevent activation of HSCs, and assists to enhance the antifibrotic pharmaceutical effect of Ssb1. This work suggests that TGF-β1 is responsible for the diminished CAT in liver fibrosis, and our designed $MnO_2$@PLGA/Ssb1 nanosystem displays enhanced antifibrotic efficiency through removing excess $H_2O_2$ and hypoxic stress, which may be a promising therapeutic approach for liver fibrosis treatment.

[1] School of Pharmacy, Zhejiang Chinese Medical University, 310053 Hangzhou, P. R. China. [2] Department of Science and Education, The First Affiliated Hospital of Guiyang University of Chinese Medicine, 550001 Guiyang, China. [3] Department of Gastroenterology, The First Affiliated Hospital, Zhejiang University School of Medicine, 310003 Hangzhou, China. [4] These authors contributed equally: Mengyun Peng, Meiyu Shao. ✉email: caogang33@163.com

Liver fibrosis has been a major global health burden with increased alcohol-use disorders and viral hepatitis infections[1]. Nowadays therapeutics are mainly focused on inhibition of hepatic stellate cells (HSCs) activation and removing excess deposited collagen, which are temporary palliative with a risk of recurrence[2]. Liver hypoxia noticeably occurs more frequently with excessive alcohol/drug consumption or liver injury and triggers liver diseases, especially liver fibrosis[3,4]. Hypoxia is reportedly associated with increased cellular oxidative stress (OS) and the upregulation of hypoxia-induced factors (HIFs) and transforming growth factor β1 (TGF-β1), which remarkably influence the progress of fibrosis[5–9]. During hypoxia, superoxide is generated at the $Q_i$ or $Q_o$ sites of complex III and is then rapidly converted into hydrogen peroxide ($H_2O_2$) through superoxide dismutase (SOD)[9–11]. Hypoxia-induced $H_2O_2$ acts as an important messenger in HIF-1α stabilization and β-catenin regulation and modulates hedgehog pathway through HIF-1α[3,12]. Meanwhile, excess $H_2O_2$ can reportedly activate macrophages and is involved in the activation of HSCs through the aquaporin 3 (AQP3) transporter[13]. In other words, hypoxia and excess $H_2O_2$ together constitute a pro-fibrotic environment, thereby accelerating the development of liver fibrosis[14].

Catalase (CAT) is the main enzyme that decomposes excess $H_2O_2$ into oxygen and $H_2O$, and presents the highest activity in mammalian liver and erythrocytes[15]. Decreased CAT activity has been widely observed and reported in many clinical liver diseases, including liver fibrosis[16], nonalcoholic steatohepatitis (NASH)[17], and hepatocellular carcinoma[4]. Loss of CAT activity is an important part of liver diseases and is usually accompanied with $H_2O_2$ accumulation rather than oxygen generation, thereby further promoting the liver hypoxia and OS[18,19]. Overexpressed CAT has been demonstrated to ameliorate liver fibrosis by inhibiting HSC activation[20,21], but not elicit sufficient attention because of complex operation and undefined mechanism. Nanozymes that mimic CAT, such as Prussian blue (PB)[22], cerium oxide (CeO₂)[23], and molybdenum disulfide (MoS₂)[24], have been utilized to scavenge $H_2O_2$ and produce oxygen for liver fibrosis therapy. CeO₂ could reduce lipid peroxidation and promote liver regeneration through diminishing oxidative stress[25]. However, reasons of decreased CAT expression in fibrotic liver were still unclear. Interactions among hypoxia, $H_2O_2$, and diminished CAT need to be understood for the identification of enhanced antifibrosis therapeutic effects.

Saikosaponin b1 (Ssb1) is one of the major active components of radix bupleuri, which is a widely used traditional Chinese medicine and is helpful in clinical liver deseases[26,27]. Saikosaponins have been demonstrated to reduce the activation of HSCs and protect liver cells from injury[28,29]. In the current study, we explored clinical specimens obtained from cirrhosis patients and a mouse model of liver fibrosis to demonstrate upregulated HIF-1α, decreased CAT activity, and $H_2O_2$ accumulation in fibrotic regions. Through RNA sequencing of mouse liver, we found that CAT decrease under hypoxia was closely related with TGF-β1. We further verified that hypoxia could induce $H_2O_2$ generation and TGF-β1 expression in HSCs, illustrating that hypoxia was associated with CAT activity through TGF-β1. Then, to ameliorate the hypoxia and OS in fibrotic liver and enhance antifibrosis efficiency, we developed a multifunctional nanodrug combining CAT-like $MnO_2$ and antifibrosis Ssb1. $MnO_2$ catalyzed excess $H_2O_2$ decomposition and supplied oxygen to the hypoxic liver, which could assist the antifibrotic effect of Ssb1. As expected, results showed that the improvement in hypoxia indeed facilitated the recovery of liver fibrosis and contributed to normalized liver oxygen and redox balance.

## Results

**Hypoxia was associated with decreased CAT activity in liver fibrosis.** Liver hypoxia reportedly increases reactive oxygen species (ROS) generation, and the resulting $H_2O_2$ is supposed to be decompose through CAT to maintain redox balance[12]. In this work, we examined the hypoxia and decreased CAT in clinical specimens from cirrhosis patients. As shown in Fig. 1a, normal and cirrhosis tissues were stained with Sirius red, and results revealed that continuously increased collagen accumulation was observed in cirrhosis liver. Collagen I, α-smooth muscle actin (α-SMA), HIF-1α, and CAT expression of normal and cirrhosis tissues were then subjected to immunofluorescence staining. Compared with normal tissues, Collagen I, α-SMA, and HIF-1α were highly overexpressed in cirrhosis tissues, and CAT expression displayed dark areas, indicating that liver fibrosis was accompanied with hypoxia and decreased CAT expression.

We further investigated the liver hypoxia/$H_2O_2$ change in an animal liver fibrosis model and treated Balb/c mice with $CCl_4$ for 5 weeks (W) and 8 weeks to mimic different degrees of hepatic fibrosis. As shown in Fig. 1b, c, it was found that compared with $CCl_4$-treated 5 W mice, increased hypoxia signal was observed with decreased CAT expression in $CCl_4$-treated 8 W mice. We then detected the $H_2O_2$ concentration of liver tissues (Fig. 1d). As expected, $H_2O_2$ concentration in $CCl_4$-treated 8 W mice was more than two-fold that of normal mice due to the glutathione (GSH) deletion and CAT deficiency[30,31]. As $H_2O_2$ has been demonstrated to play important roles in fibrosis generation, CAT catalytic activity is critical in fibrosis recovery[32]. For further validation, LX-2 and HSC-T6 were treated with 5% $O_2$ for 12 and 24 h and significantly decreased of CAT expression was detected in LX-2 and HSC-T6 cells (Fig. 2d). These results illustrated the fact that hypoxia was responsible for CAT decrease and resulted in $H_2O_2$ accumulation.

**Hypoxia downregulated CAT expression through TGF-β1.** To further confirm the interaction between hypoxia and CAT in liver fibrosis, the liver of $CCl_4$-treated 8 W mice and normal mice were subjected to RNA transcriptomics analysis. Downregulation of CAT was detected in $CCl_4$-treated mice with statistical significance, and the decrease in CAT involved GO terms such as oxidoreductase activity, oxidation-reduction process, and response to hypoxia (Fig. 2a). In the GO term "response to hypoxia", we found that hypoxia-correlated *Egln3*, *Hif-1a*, and fibrosis-associated genes (*Mmp2*, *Mmp14*, *Tgfb1*, *Tgfb2*, and *Smad3*) were observably upregulated in fibrotic mice, but cellular key antioxidant enzymes, such as *CAT* and *Sod2* were decreased (Fig. 2b). Protein–protein interaction was analyzed based on STRING database, and the *CAT*-correlated network was visualized by Cystoscope (v.3.8.2). The master profibrogenic cytokine TGF-β1 was directly related to CAT (Fig. 2c).

In previous works, TGF-β1 has been demonstrated to strongly suppress CAT mRNA by activating of Smad3 in airway smooth muscle cells[33,34]. TGF-β1/Smad3 pathway regulated expression of Foxo3a (forkhead box type O3a) and subsequent CAT mRNA has been further proved in cardiac fibrosis[35]. To confirm that TGF-β1 was responsible for CAT regulation, HSCs were sequentially incubated with hypoxia and TGF-β1 to verify the factor influencing CAT expression (Fig. 2d and Supplementary Fig. 1a, b). In HSC-T6 and LX-2 cells, hypoxia was observed to induce significantly increased of TGF-β1 and reduce CAT expression. TGF-β1-treated cells showed efficient suppression of CAT, indicating that hypoxia-induced TGF-β1 upregulation and thus decreased CAT expression. Smad3-specific inhibitor, SIS3, was used to inhibit TGF-β1 induced Smad3 activation (Supplementary Fig. 1c). As shown in Fig. 2e,

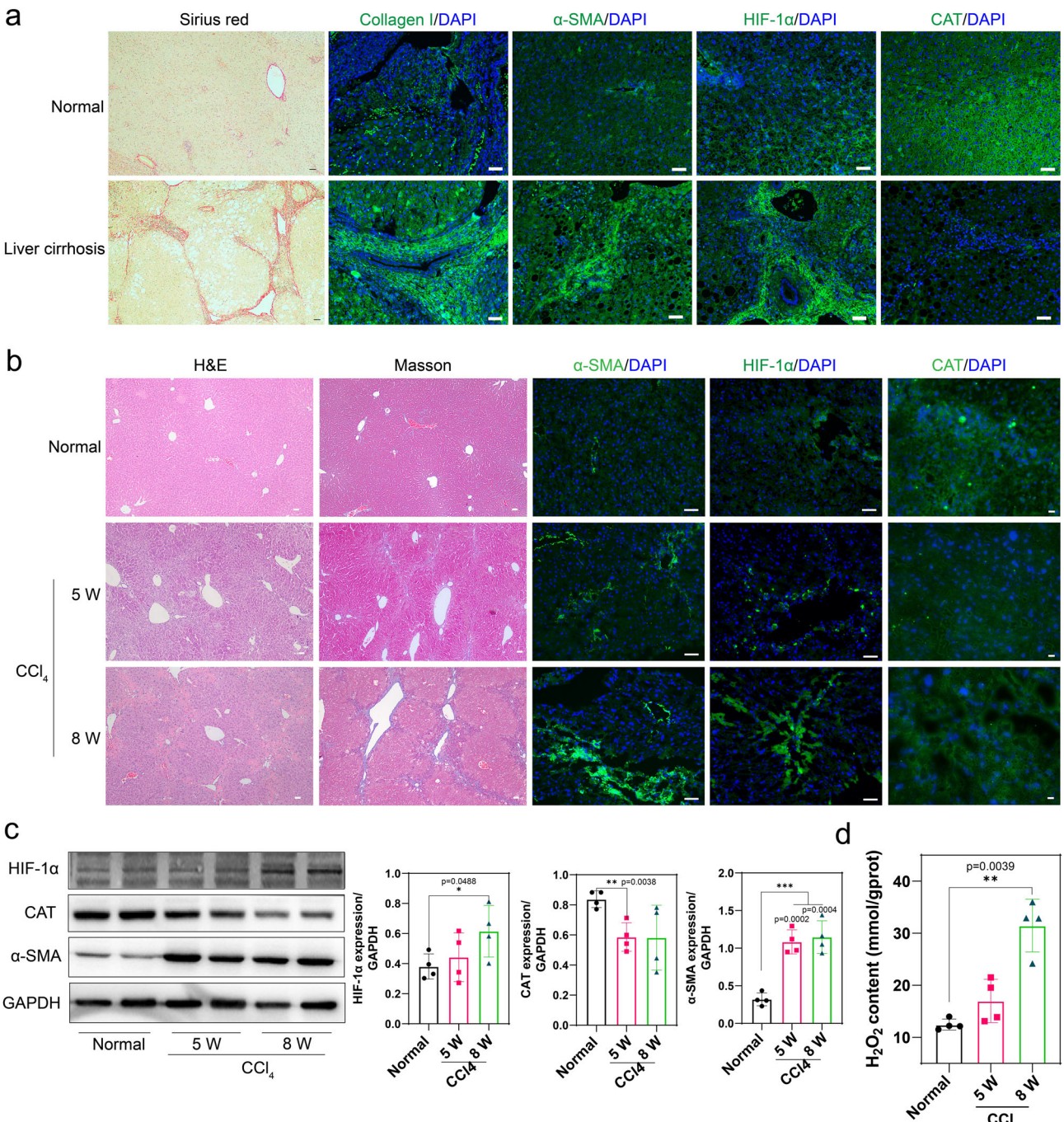

**Fig. 1 Hypoxia was associated with decreased CAT activity in liver fibrosis. a** Representative images of Sirius red staining (scale bar, 50 μm) and immunostaining for Collagen I, a-SMA, HIF-1α, and CAT (scale bar, 50 μm) in clinical samples of indicated patients. **b** Representative haemotoxylin and eosin (H&E) and Masson staining (scale bar, 50 μm), immunostaining for a-SMA, HIF-1α (scale bar, 50 μm), CAT (scale bar, 10 μm) of liver tissue sections from normal mice, and mice treated with CCl₄ for 5 or 8 W. Scale bar: 50 μm. **c** Representative western blot (WB) analysis ($n = 4$, Mean ± S.D.) of hepatic HIF-1α, CAT, α-SMA expression from normal mice, and mice treated with CCl₄ for 5 and 8 W ($n = 4$, Mean ± S.D.). **d** H₂O₂ content in the liver of normal mice and mice treated with CCl₄ for 5 or 8 W ($n = 4$, Mean ± S.D.). $*p < 0.05$, $**p < 0.01$, and $***p < 0.001$ (unpaired Student's $t$ test) versus normal mice.

TGF-β1 treated HSCs were effectively activated and down regulated expression of Foxo3a and CAT (Supplementary Fig. 1d). This process was markedly blocked with SIS3 due to inhibition of Smad3 activation, and CAT mRNA expression was highly recovered after SIS3 treated active HSCs (Fig. 2f). Furthermore, Nrf2 (nuclear factor (erythroid-derived 2)-like 2) was another key regulator of CAT[36]. In hypoxia and TGF-β1 activated HSCs, Nrf2 expression was decreased (Supplementary Fig. 2a, b). Nrf2 was further silenced through Nrf2 shRNA, and decreased CAT was expressed compared

with control group, indicating that Nrf2 perturbation contributed to CAT regulation that respond to both hypoxia and TGF-β1 induction (Supplementary Fig. 2c–e). Therefore, TGF-β1 triggered by hypoxia could decrease CAT expression through inhibition of Foxo3a and Nrf2, which induced severe H₂O₂ accumulation in liver and could be an important target for antifibrosis therapy.

Afterwards, we examined hypoxia-induced H₂O₂ generation by co-staining 2,7-dichlorofluorescein diacetate (DCFH-DA) and MitoTracker Deep Red through confocal laser scanning

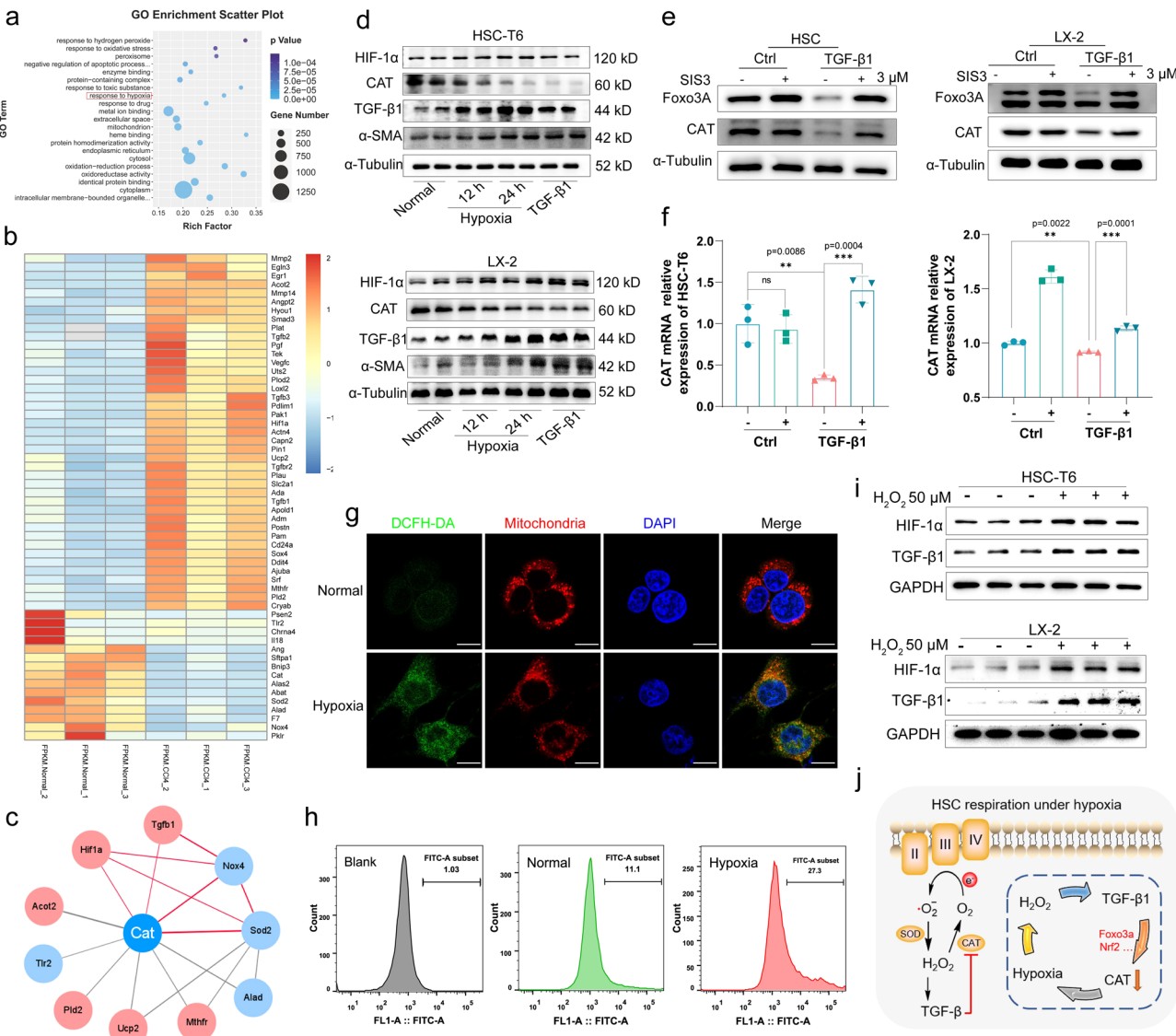

**Fig. 2 Hypoxia downregulated CAT expression through TGF-β1. a** GO enrichment analysis of GO terms including CAT and HIF-1α in the liver of fibrotic mice. **b** Heatmap for showing the differently expressed genes related to "response to hypoxia" GO term. **c** Functional-association networks of the CAT-correlated gene, the analysis was based on STRING database for the retrieval of protein-protein interaction. **d** Representative WB analyze of HIF-1α, TGF-β1, CAT, and a-SMA under different treatment on HSC-T6 and LX-2 cells. **e** WB analyze of Foxo3a and CAT of HSC-T6 and LX-2 stimulated with TGF-β1 10 ng/mL for 24 h in presence or absence of SIS3. **f** RT-qPCR analysis of CAT mRNA expression in HSC-T6 and LX-2 ($n = 3$, Mean ± S.D., *$p < 0.05$, **$p < 0.01$; ***$p < 0.001$ by Student's $t$ test). **g** CLSM images and mitochondria in HSC-T6 cells. Representative confocal images of Mito Tracker Deep Red FM and DCFH-DA staining in HSC-T6 under 5% $O_2$ hypoxia treatment. Scale bar, 10 μm. **h** Intracellular ROS levels analyzed by flow-cytometry analysis. **i** TGF-β1 expression in HSC-T6 and LX-2 induced by $H_2O_2$. **j** Schematic illustration of the hypoxia-regulating mechanism in HSCs.

microscopy (CLSM) (Fig. 2g), and then quantified it by flow cytometry (FCM) analysis (Fig. 2h). The intensity of DCF was weak under normoxia. However, upon hypoxia treatment, DCF fluorescence increased and showed good co-localization with the mitochondrial region. As a kind of ROS, $H_2O_2$ could increase the OS and stimulate HSCs transition into activated myofibroblast[3]. Increased concentration of $H_2O_2$ (50 μM) was then incubated with HSC-T6 cells and LX-2 cells, and $H_2O_2$-induced TGF-β1 expression was examined in Fig. 2i and Supplementary Fig 1e, f. Results indicated one possible pathway between hypoxia and CAT, as illustrated in Fig. 2j. In a typical procedure, liver hypoxia could increase $H_2O_2$ generation through mitochondrial respiration, and the resulting OS activated TGF-β1 expression. Over-expressed TGF-β1 could regulate CAT expression through downregulation of Foxo3a and Nrf2, which in turn protected

$H_2O_2$ from decomposition. This vicious cycle provided chronic stimulation of HSCs and was a main barrier in antifibrosis drug therapy.

## Syntheses and characteristics of CAT-like MnO$_2$@PLGA/Ssb1 nanodrug.
Liver $H_2O_2$ accumulation provided chronic stimulation of HSCs and destroyed the normal liver microenvironment. Therefore, we introduced a CAT-like compound MnO$_2$ to decompose excess $H_2O_2$ and produce $O_2$, which could relieve the OS and tissue hypoxia[37,38]. In the present work, we designed a nanodrug with biodegradable poly (lactic-co-glycolic acid) (PLGA) nanoparticles (NPs) loading a hydrophobic antifibrotic drug Ssb1 and then coated it with MnO$_2$ shells (MnO$_2$@PLGA/Ssb1 NPs) (Fig. 3a)[39,40]. After injection and body circulation, nanosized

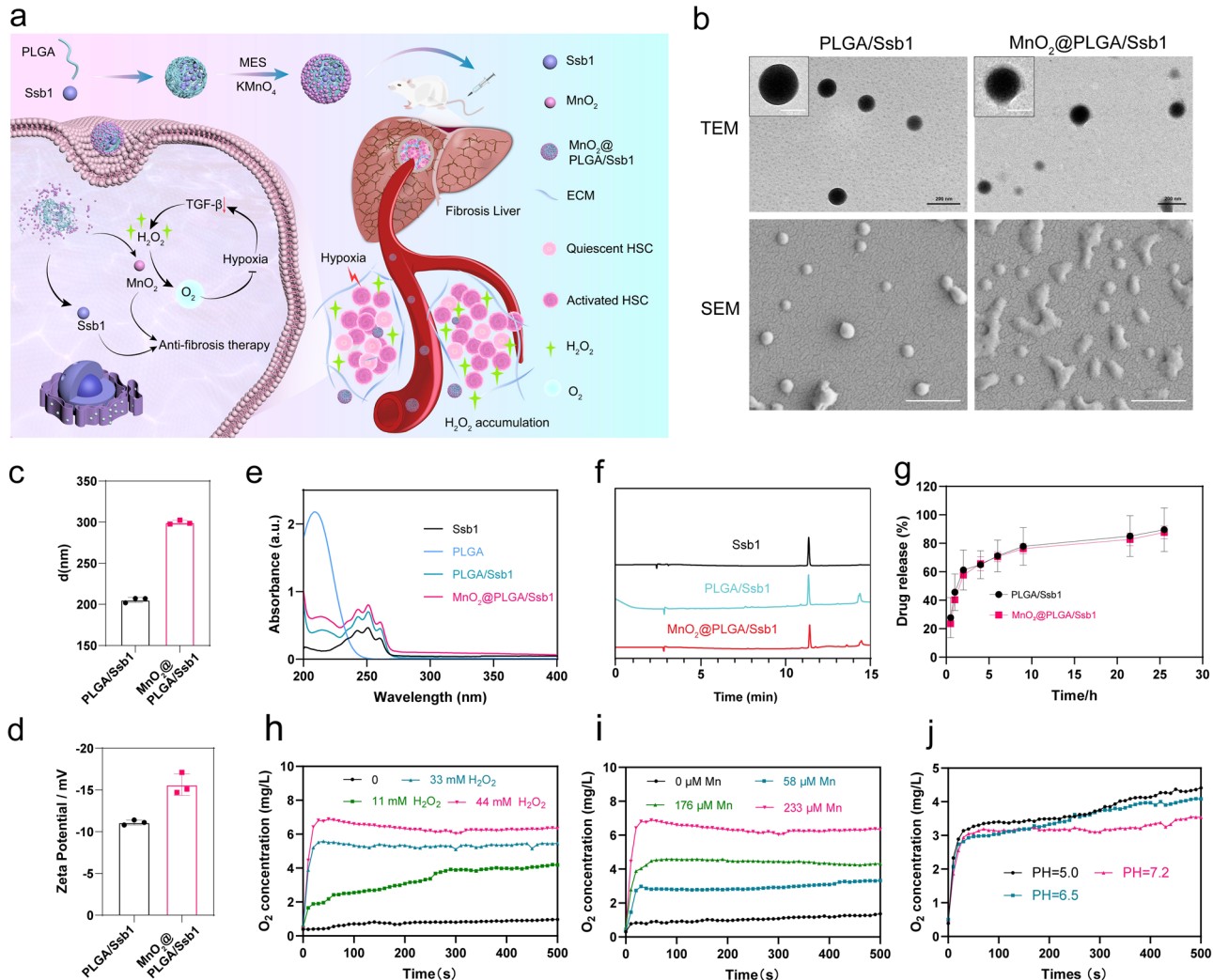

**Fig. 3 MnO$_2$@PLGA/Ssb1 nanosystem design and characterization. a** Illustration of catalase-like MnO$_2$@PLGA/Ssb1 nanosystem for liver-fibrosis therapy via modulating the fibrotic microenvironment. **b** Representative scanning electron microscopy (SEM) and transmission electron microscopy (TEM) images of different nanoparticles. Scale bar, 100 nm. **c** Average size distribution of different nanoparticles ($n = 3$, Mean ± S.D.). **d** Zeta potential of different nanoparticles ($n = 3$, Mean ± S.D.). **e** UV-vis spectra of Ssb1, PLGA, PLGA/Ssb1, and MnO$_2$@PLGA/Ssb1. **f** HPLC spectra of Ssb1 loaded in PLGA/Ssb1 and MnO$_2$@PLGA/Ssb1 nanoparticles. **g** Drug-release rate of PLGA/Ssb1 and MnO$_2$@PLGA/Ssb1 nanoparticles ($n = 3$, Mean ± S.D.). **h** O$_2$ generation of different H$_2$O$_2$ amounts after adding MnO$_2$@PLGA/Ssb1 (Mn concentration = 233 μM). **i** O$_2$ generation of different MnO$_2$@PLGA/Ssb1 with 44 mM H$_2$O$_2$. **j** O$_2$ generation of MnO$_2$@PLGA/Ssb1 (Mn concentration = 117 μM) and H$_2$O$_2$ (22 mM) in buffers of different pH values.

MnO$_2$@PLGA/Ssb1 NPs were accumulated in fibrotic liver with enhanced vascular permeability[41]. Cellular H$_2$O$_2$ was further catalyzed into O$_2$, which ameliorated hypoxic stimulation and synergistically enhanced the therapeutic efficiency of released Ssb1.

Ssb1-loaded PLGA NPs (PLGA/Ssb1 NPs) were synthesized according to a previous researches[42]. And MnO$_2$ was grown on the surface of PLGA/Ssb1 NPs through the redox of potassium permanganate (KMnO$_4$) to construct MnO$_2$@PLGA/Ssb1 NPs[43]. Microscopy images revealed the successful coating of MnO$_2$ onto PLGA/Ssb1 NPs, especially in scanning electron microscopy (SEM) images. The successful coating of MnO$_2$ on PLGA provided a hard support to maintain the spherical morphology (Fig. 3b). Supplementary Fig. 3a showed the element mapping of Mn and C in MnO$_2$@PLGA/Ssb1 NPs. C element was concentrated in a portion of the selection area and Mn element was dispersed across the selection area. The merged mapping image further supported that MnO$_2$ successfully formed on the surface of NPs. Average hydrodynamic diameters and zeta potentials of PLGA/Ssb1 and MnO$_2$@PLGA/Ssb1 NPs were measured by dynamic light scattering (DLS), The size and potential value of MnO$_2$@PLGA/Ssb1 were higher than those of PLGA/Ssb1, which could be attributed to the MnO$_2$ coating on PLGA/Ssb1 NPs and afforded good in vitro and in vivo stability (Fig. 3c, d)[44,45].

Ssb1 loading in NPs was analyzed by UV-visible (UV-vis) absorption spectroscopy. A signature triple peak of Ssb1 within 270–400 nm was observed, and the absorption peak of PLGA at 210 nm in PLGA/Ssb1 and MnO$_2$@PLGA/Ssb1 NPs was also found to be consistent with the designed structure (Fig. 3e). Accurate determination of Ssb1 content in PLGA/Ssb1 and MnO$_2$@PLGA/Ssb1 NPs was applied through HPLC. The standard curve of Ssb1 was detected (Supplementary Fig. 3b), and the encapsulation efficiency of Ssb1 in PLGA/Ssb1 and MnO$_2$@PLGA/Ssb1 reached up to 65% and 52.5% (Fig. 3f). We then investigated the in vitro drug release of NPs in pH 7.4. NPs showed similar release rate and released approximately 85% of the encapsulated Ssb1 in pH 7.4 phosphate-buffered saline (PBS) within 28 h (Fig. 3g).

**In vitro study of $H_2O_2$ splitting and oxygen generation**. $MnO_2$ is known to mimic CAT-like enzymes for catalyzing $H_2O_2$ into $H_2O$ and oxygen, and nanosized $MnO_2$@PLGA/Ssb1 is supposed to show higher catalytic efficiency due to its high surface-to-volume ratio[46]. The $H_2O_2$-splitting capability of $MnO_2$@PLGA/Ssb1 was measured with a dissolved-oxygen meter. With increased concentration of $H_2O_2$ from 0 to 44 mM, $MnO_2$-based nanoplatform ($MnO_2$@PLGA/Ssb1) yielded satisfactory effect of $O_2$ production. As shown in Fig. 3h, with increased $H_2O_2$ concentration, the catalysis reaction was apparently enhanced and the dissolved $O_2$ was rapidly increased from 0.5 mg/L to 6.9 mg/L within 50 s. Afterwards, the oxygen-generation efficiency of $MnO_2$@PLGA/Ssb1 at different concentrations was also examined with 44 mM $H_2O_2$. Increased $O_2$ generation was correlated with the $MnO_2$@PLGA/Ssb1 catalyst content, and results indicated the catalytic capacity of $MnO_2$@PLGA/Ssb1 nanodrug against $H_2O_2$ (Fig. 3i). We further investigated the influence of pH value on $O_2$ generation rate (Fig. 3j). In pH 5.0 and pH 6.5 buffers, $O_2$ generation was slightly higher than in pH 7.2 buffer, which meant higher catalytic efficiency of $MnO_2$ under weak acid condition.

**Cellular antifibrotic efficiency of $MnO_2$@PLGA/Ssb1**. Ssb1 is a hydrophobic antifibrotic compound from bupleurum herb[47]. In the current work, its antifibrotic effect was demonstrated in TGF-β1-activated HSC-T6/LX-2 cells (Fig. 4a and Supplementary Fig. 4a–c). As an efficient profibrotic cytokine, TGF-β1 was applied to activate HSCs into myofibroblasts and upregulated α-SMA expression[48]. In all, 15 μM Ssb1 could significantly inhibit the expression of α-SMA and Collagen I in TGF-β1-stimulated HSCs. Additionally, cytotoxicity of Ssb1 demonstrated that activated HSCs were more sensitive to high-concentration Ssb1 than quiescent HSCs. We further detected the expression of apoptotic protein caspase 3, results showed that cleaved caspase 3 was increased in a gradient-dependent manner, indicating that Ssb1 could induce apoptosis of activated HSCs.

The antifibrotic efficiency of $MnO_2$@PLGA/Ssb1 was investigated in TGF-β1-activated HSC-T6/LX-2 cells. Mitochondrial reactive oxygen species (ROS) increased in activated HSCs, thereby inducing stable expression of HIF-1α[49]. As displayed in Fig. 4b, c, α-SMA and HIF-1α were significantly upregulated in TGF-β1-treated cells. In other groups, Ssb1, $MnO_2$, and Ssb1-loaded PLGA/Ssb1 NPs inhibited α-SMA and HIF-1α expression in activated HSCs, and $MnO_2$@PLGA/Ssb1-treated cells showed the best antifibrosis efficiency due to the synergistic effect of $MnO_2$ and Ssb1 (Supplementary Fig. 4d). $MnO_2$ diminished hypoxia and ROS stimulation to HSCs, thereby enhancing the antifibrosis effect of Ssb1. The expression of α-SMA and HIF-1α were further evaluated by immunofluorescent-staining method, and results similar to WB were obtained (Fig. 4d, e)

**$MnO_2$@PLGA/Ssb1 could reduce hypoxia-induced $H_2O_2$ and TGF-β1**. We applied 24 h of hypoxia (5% $O_2$) treatment to induce HSC-T6 and LX-2 cell activation due to the results in Fig. 2d. Under hypoxia, mitochondrial $H_2O_2$ generation increased in HSCs and induced TGF-β1 expression, which reprogrammed HSCs into myofibroblasts. In Fig. 4f and Supplementary Fig. 5a, cellular ROS was firstly detected by CLSM and FCM analyses. Compared with the hypoxia-treated group, $MnO_2$@PLGA/Ssb1-incubated cells presented decreased DCF fluorescence because of the decomposition of $H_2O_2$, indicating that $MnO_2$@PLGA/Ssb1 could reduce hypoxia-induced ROS and avoid OS of HSCs. Then, the TGF-β1 expression of hypoxia and different material-treated HSCs were detected. As shown in Fig. 4g, h, $MnO_2$@PLGA/Ssb1 could efficiently decrease TGF-β1 and HIF-1α expression in HSC-

T6 and LX-2 cells (Supplementary Fig. 4e). Results illustrated that our designed $MnO_2$@PLGA/Ssb1 nanodrug successfully reduced profibrotic factors, indicating its potential applications in anti-fibrosis therapy. Furthermore, according to the accumulation of $H_2O_2$ under hypoxia, we also detected $H_2O_2$ induced TGF-β1 expression in HSC-T6 (Supplementary Fig. 5b). $MnO_2$@PLGA/Ssb1 treatment could efficiently decrease TGF-β1 due to the relief of OS induced by $H_2O_2$ in HSCs.

**In vivo distribution of Ssb1 and $MnO_2$ in main organs**. NPs were labeled with the near-infrared-absorbing cyanine dye 1,1-dioctadecyl-3,3,3,3-tetramethylindotricarbocyanine iodide (DiR iodide) for optical imaging (PLGA/DiR and $MnO_2$@PLGA/DiR NPs)[50]. The in vivo accumulation of PLGA/DiR and $MnO_2$@PLGA/DiR during body circulation was monitored using a small-animal fluorescence-imaging system. After injection for 8 hours, fluorescence of DiR appeared to accumulated in mice and main organs were subsequently imaged (Fig. 5a, b). Compared with free DiR, PLGA/DiR and $MnO_2$@PLGA/DiR were mostly concentrated in the liver and lung with particle size about 200 nm. This phenomenon was consistent with previous reports that spherical particles beyond 150 nm tended to be entrapped mostly in the liver and lung after body circulation[41,51,52].

The biodistribution of Ssb1 and $MnO_2$ in main organs was further quantified by liquid chromatography mass spectrometry (LC-MS) and inductively coupled plasma mass spectrometry (ICP-MS) for dosage determination in antifibrotic therapy. As shown in Fig. 5c, Ssb1 mainly distributed in liver, lung and kidney, whereas the distribution of Ssb1 only, PLGA/Ssb1, and $MnO_2$@PLGA/Ssb1 did not show significant differences. Figure 5d shows that the manganese concentrations of $MnO_2$ and $MnO_2$@PLGA/Ssb1 were similar in main organs, and manganese element was specifically concentrated in the liver and kidney. The quantitative results of Ssb1 and Mn were coincided with those of DiR labeled in vivo imaging and thus could guide the following therapeutic dosage.

**$MnO_2$ synergistically enhanced the antifibrotic efficiency of Ssb1**. Fibrotic mice were established with 40% $CCl_4$ injection for 5 W and then analyzed with WB, hematoxylin & eosin (H&E) and Masson's trichrome staining (Supplementary Fig. 6a, b). Based on the biodistribution of Ssb1 and $MnO_2$ in mouse liver, therapeutic dosage was calculated and intravenously injected once a week (Fig. 6a). After treatment for three weeks, mice were sacrificed for the determination of antifibrosis indices. Ssb1-treated mice were performed with RNA-sequencing, and differential expressed genes were mostly enriched in the "ECM-receptor interaction" KEGG pathway. In the heatmap of this pathway, it was obvious that collagen genes, such as Col1a1, Col1a2, Col4a1, Col4a2, Col4a5, Col6a1, Col6a2, and Col6a3 were significantly decreased in Ssb1 group, indicating that Ssb1 could downregulate collagen expression in fibrotic liver (Supplementary Fig. 7).

As shown in Fig. 6b, $H_2O_2$ was highly accumulated in untreated fibrotic liver and decreased in the $MnO_2$, PLGA/Ssb1, and $MnO_2$@PLGA/Ssb1-treated groups. The decrease of in $H_2O_2$ content reduced the OS of the liver, thereby avoiding additional activation of quiescent HSCs[53]. The expression of HIF-1α and TGF-β1 in $MnO_2$@PLGA/Ssb1-treated mice was significantly reduced as expected because of the decomposition of $H_2O_2$ and $O_2$ generation (Fig. 6c and Supplementary Fig. 6c). Moreover, due to the synergistic coordination of $MnO_2$ and Ssb1, fibrotic proteins α-SMA and Collagen I were also decreased in $MnO_2$, PLGA/Ssb1, $MnO_2$@PLGA/Ssb1-treated mice. Liver tissues were sliced and immunofluorescence stained to monitor the tissue

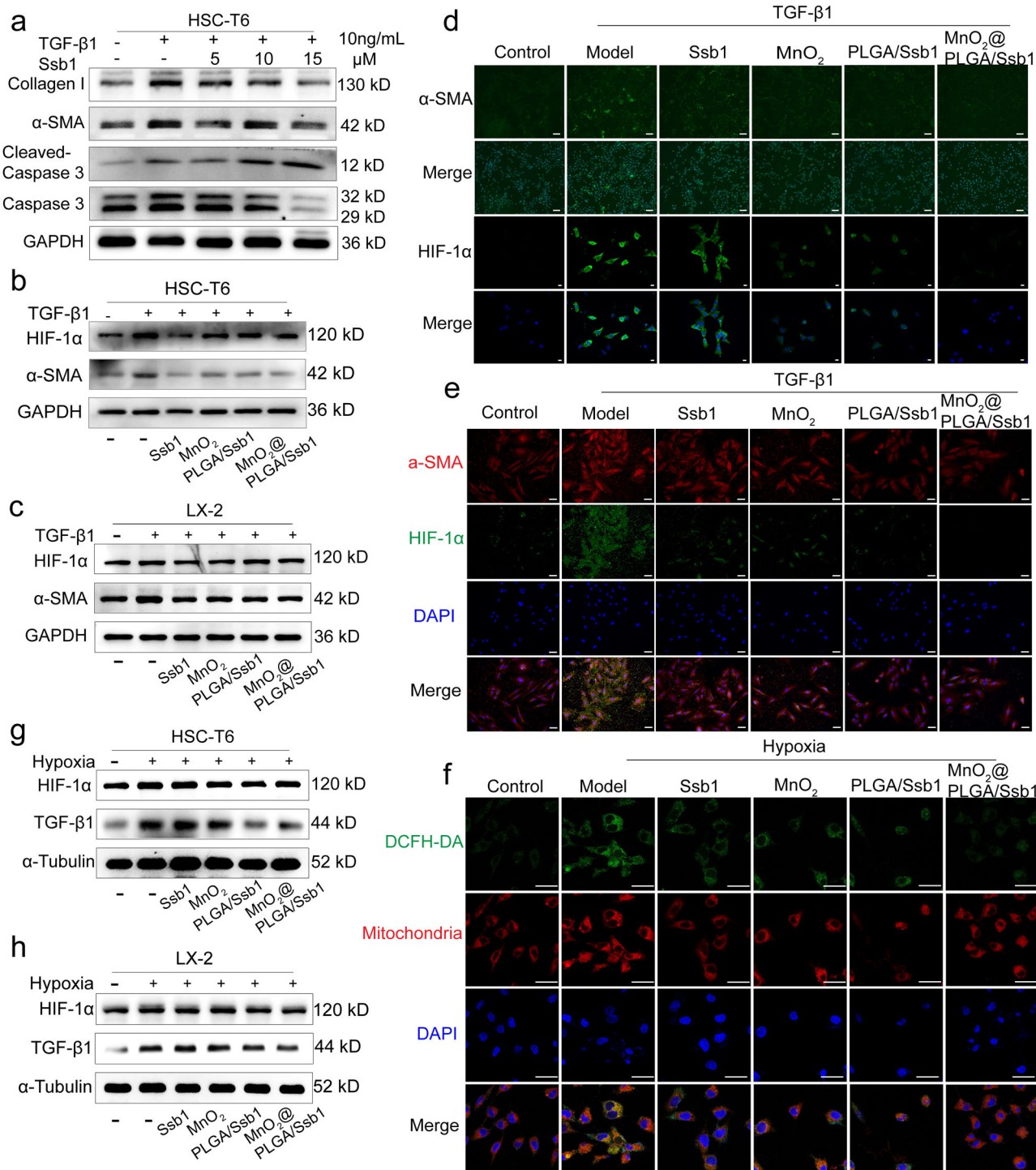

**Fig. 4 Anti-fibrotic efficacy. a** HSC-T6 cells were exposed to TGF-β1 (10 ng/mL) and treated with Ssb1 (0–15 μM) for 24 h. Expression of Collagen I, α-SMA, and Caspase 3 were determined by western blot assay. **b–e** Cellular antifibrotic efficiency of MnO₂@PLGA/Ssb1. α-SMA and HIF-1α expression of the HSC-T6 (**b**) and LX-2 cells (**c**) with different treatments challenged by TGF-β1 were assayed by WB. The expressions levels of α-SMA and HIF-1α of HSC-T6 (**d**) and LX-2 cells (**e**) were evaluated by immunofluorescent staining. **f** Cellular ROS was detected by CLSM in HSC-T6 cells with different treatments under 5% O₂ hypoxia. Scale bar, 50 μm. **g, h** MnO₂@PLGA/Ssb1 could reduce hypoxia-induced H₂O₂ and TGF-β1. The HIF-1α and TGF-β1 expression levels of the HSC-T6 (**g**) and LX-2 (**h**) with different treatments challenged by hypoxia were assayed by WB.

expression of HIF-1α, CAT, and α-SMA (Fig. 6d and Supplementary Fig. 6d). Compared with fibrotic mice and Ssb1 therapy, hypoxia was efficiently ameliorated in MnO₂@PLGA/Ssb1-treated mice, and inducing the downregulation of HIF-1α and α-SMA in liver. TUNEL staining was performed with co-staining

of α-SMA. In Fig. 6e, nanosized PLGA/Ssb1 and MnO₂@PLGA/Ssb1 displayed more efficient apoptosis than Ssb1, indicating higher utilization ratio of Ssb1 in nanoforms. The H&E and Masson-stained liver slices were further investigated, and results showed that liver injury induced conspicuous inflammatory cell

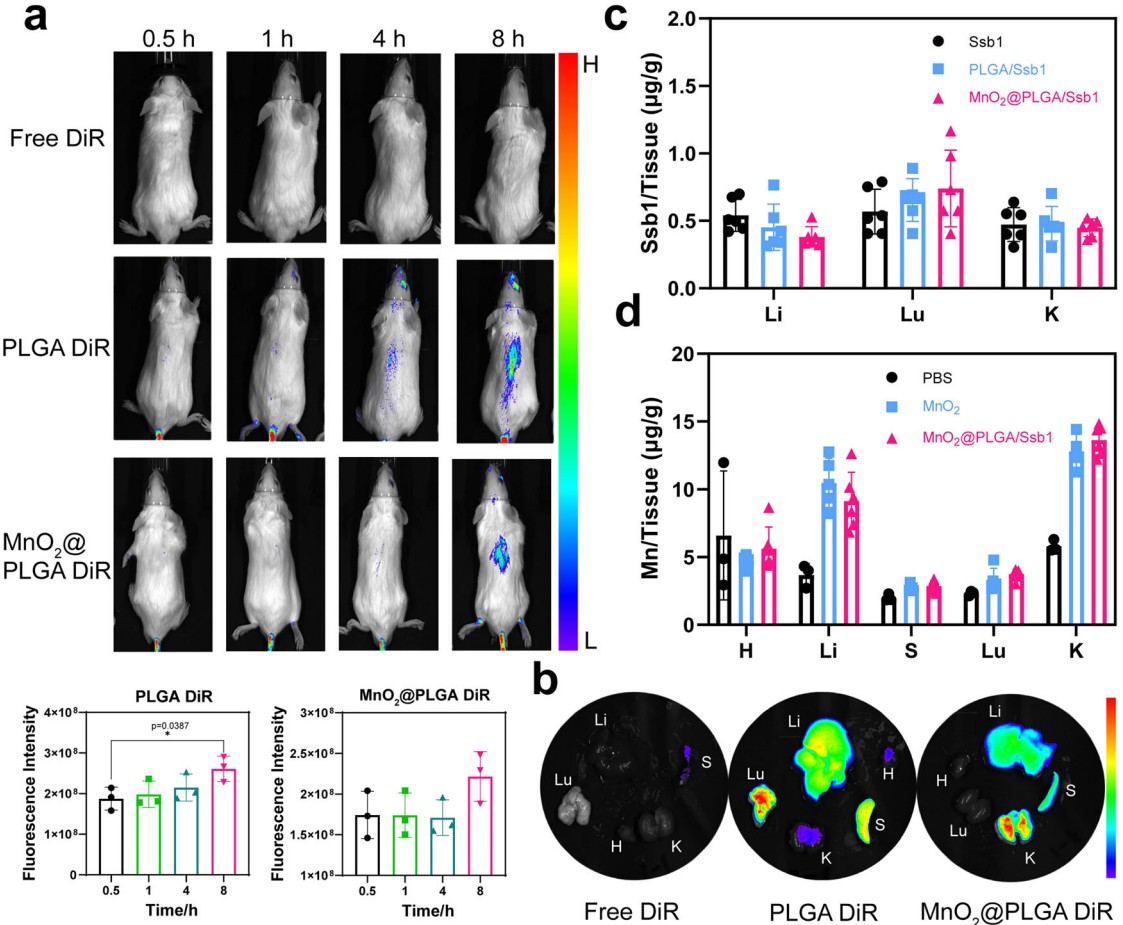

**Fig. 5 Biodistribution of Ssb1 and MnO$_2$ in main organs in vivo. a** Representative in vivo fluorescence images of mice receiving DiR-loaded nanoparticle by i.v. injection. ($n = 3$, Mean ± S.D., *$p < 0.05$ by Student's $t$ test). **b** Ex fluorescence imaging of liver and other important organs, 8 h post-injection of DiR-loaded nanoparticles (H, Li, S, Lu, and Ki represent heart, liver, spleen, lung, and kidney). **c** Quantification of Ssb1 in tissue homogenates ($n = 6$, Mean ± S.D.). **d** Biodistribution of Mn element after the intravenous injection with MnO$_2$ and MnO$_2$@PLGA/Ssb1 nanoparticles ($n = 5$, Mean ± S.D.).

aggregation and crisscross collagen fibers in untreated mice, this phenomenon was visibly alleviated in MnO$_2$@PLGA/Ssb1-treated mice for fibrosis recovery (Fig. 6f).

Additional hematological examination (Fig. 6g) revealed that liver function indices, including TBIL, ALT, and AST were significantly recovered in the MnO$_2$@PLGA/Ssb1 group, indicating that MnO$_2$@PLGA/Ssb1 was truly efficient for antifibrosis therapy. Four indicators (HA, LN, PIIINP, and IV-C) of mice serum were also measured by enzyme-linked immunoSorbent assay (ELISA) to further conform with the clinical test indices. As shown in Fig. 6h, serum levels of laminin (LN), collagen type IV (CIV), and PIIINP decreased to a greater extent in fibrotic mice injected with MnO$_2$@PLGA/Ssb1 than in those injected with PBS. All these results indicated that the combination of MnO$_2$ and Ssb1 did achieve enhanced anti-fibrosis efficiency.

**In vitro and in vivo biosafety assay.** Biosafety of Ssb1, PLGA/Ssb1, and MnO$_2$@PLGA/Ssb1 was investigated both in vitro and in vivo to ensure the safety of our materials. Quiescent HSC-T6 and LX-2 cells were initially incubated with different concentrations of Ssb1, PLGA/Ssb1, and MnO$_2$@PLGA/Ssb1 (Fig. 7a, b). 50 µM Ssb1 did not disturb cellular viability both in HSC-T6 and LX-2, and Ssb1-loaded PLGA/Ssb1 and MnO$_2$@PLGA/Ssb1 did not show extra burden to cell growth, indicating good biocompatibility of this antifibrotic nanodrug. Then, in vivo acute toxicity of nanodrug to Balb/c mice was investigated with three

days' continuous injection. Tissues (heart, liver, spleen, lung, and kidney) were collected and analyzed by H&E staining (Fig. 7c). As demonstrated in Fig. 5b, PLGA/Ssb1 and MnO$_2$@PLGA/Ssb1 NPs mainly accumulated in the liver and lung, and histopathological examinations of liver and lung showed no significance in inflammatory cell infiltration or tissue morphology. Hematological examination also revealed nearly no differences among blood ALT, AST, and ALP values (Fig. 7d). These results illuminated the satisfactory biosafety of utilized Ssb1, MnO$_2$, and Ssb1-loaded NPs.

## Discussion

TGF-β1 is the classical pro-fibrotic cytokine to drive HSCs activation and has been widely used for construction of in vitro fibrotic model[54]. In previous works, the influence of TGF-β1 on CAT has been explored in airway smooth muscle cells but did not get more attentions[33]. Diminished CAT activity in the liver/lung fibrosis[55] has been observed over the past decades and is reportedly responsible for the cellular redox imbalance[56]. The present study showed that liver hypoxia induces increased TGF-β1 expression, which effectively inhibited CAT expression through downregulation of Foxo3a and Nrf2 in HSCs. As the result of liver CAT inhibition, increased H$_2$O$_2$ accumulation and stabilized HIF-1α were subsequently formed in the fibrotic area. What's more, H$_2$O$_2$ and HIF-1α could further stimulate the

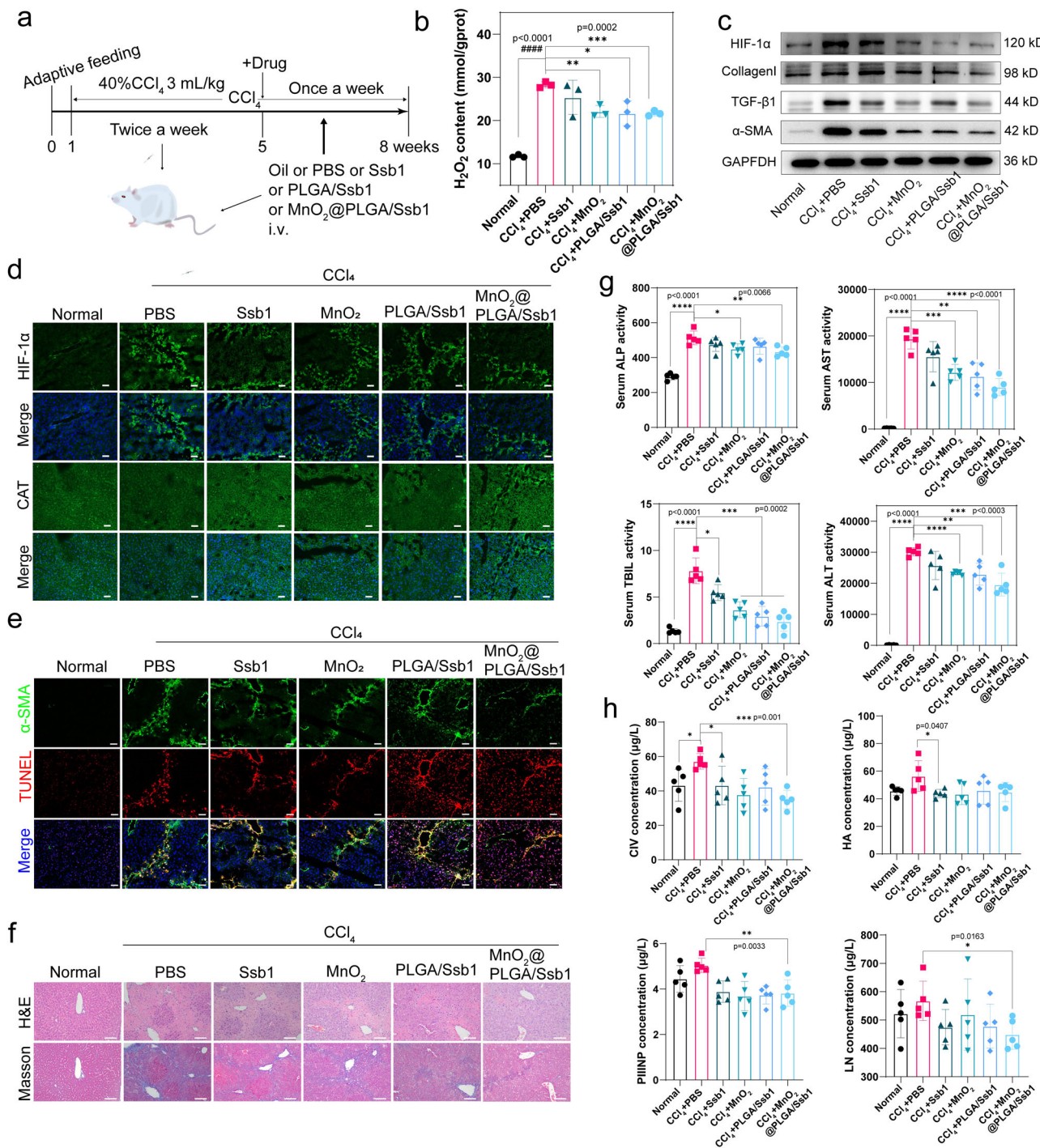

**Fig. 6 Antifibrotic activity of CAT-mimicking nanodrug in vivo. a** Overall procedure of the animal experiment (CCl₄-induced liver fibrosis). **b** $H_2O_2$ content of liver with different treatment ($n = 3$, Mean ± S.D., #model compared with normal, *other groups compared with model). **c** Hepatic HIF-1α, Collagen I, TGF-β1, and α-SMA expression levels were measured by WB. **d** Hepatic CAT, HIF-1α were evaluated by immunofluorescent staining (scale bar, 50 μm). **e** α-SMA and TUNEL staining of mouse tissue sections (scale bar, 50 μm). **f** Representative H&E and Masson-stained liver tissue sections. Blue areas in Masson-stained sections indicate collagen deposition in fibrotic liver tissues (Scale bars, 100 μm). **g** Serum levels of AST, ALT, TBIL, and ALP was measured by biochemical assays ($n = 5$, Mean ± S.D.). **h** Four indicators (HA, LN, PCIII, and IV-C) of serum liver fibrosis was measured by ELISA ($n = 5$, Mean ± S.D.). (*$p < 0.05$, **$p < 0.01$; ***$p < 0.001$ by Student's $t$ test).

upregulation of TGF-β1, which set up a vicious cycle and acted as a difficult barrier for liver fibrosis treatment[13].

We proposed that the hypoxia and OS of liver provided chronic and constant stimulations toward HSCs, thereby conferring difficulty in fibrosis recovery. In the pro-fibrotic environment, TGF-β1 diminished CAT activity and then induced

$H_2O_2$ accumulation. The excess $H_2O_2$ in turn further increased TGF-β1 and established a vicious cycle. For this reason, $H_2O_2$ was selected as the target to break the circle, and $MnO_2$ was applied to represent CAT-like activity for $H_2O_2$ decomposition. $MnO_2$ remodeled the fibrotic microenvironment by ameliorating hypoxia and OS, thereby reducing a pro-fibrotic stimulus from

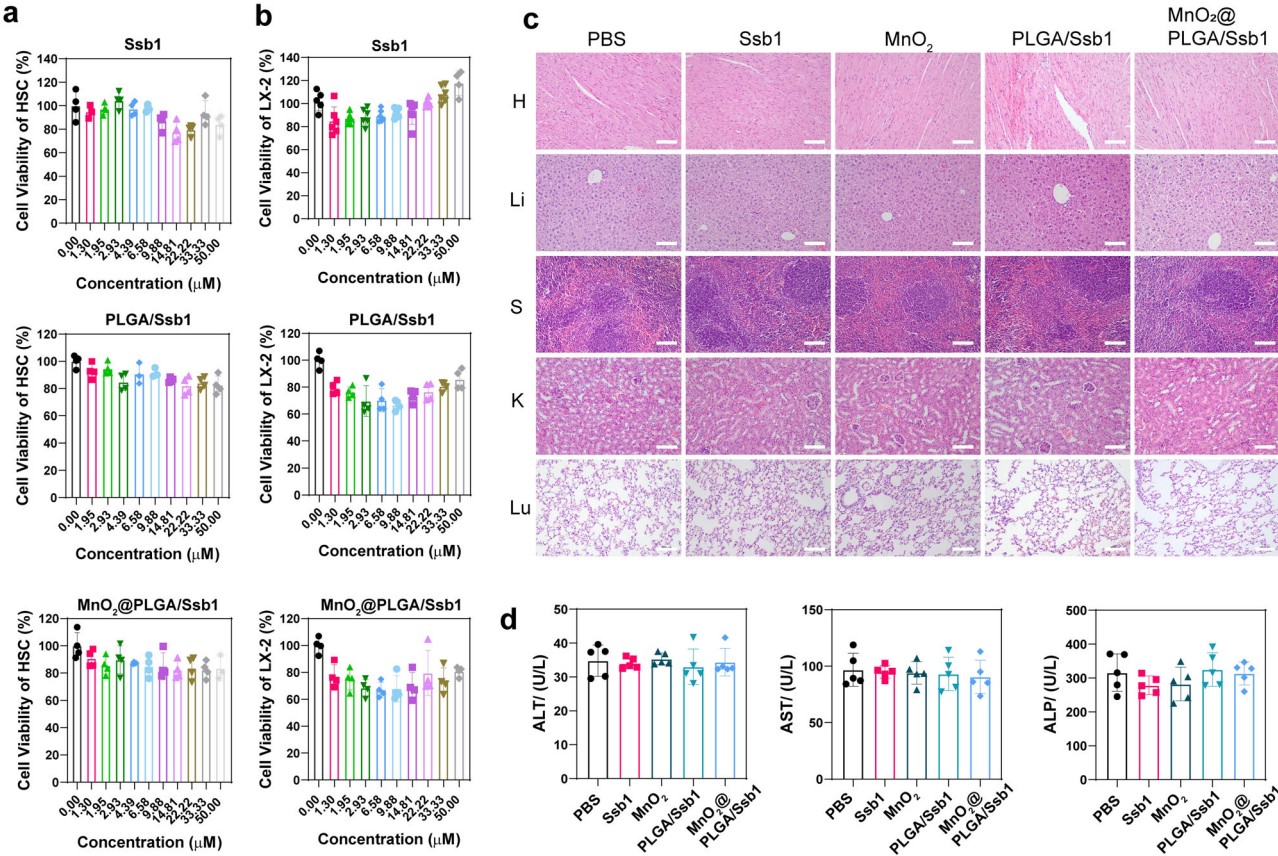

**Fig. 7 Biosafety assay of CAT-mimicking nanodrug.** Cell viability of HSCs (**a**) and LX-2 (**b**) cells with different concentrations of materials ($n = 4$, Mean ± S.D.). H&E-stained images (**c**) and blood biochemistry (**d**) and of organs including liver, spleen, kidney, heart, and lung from mice injected with 8.25 mg/kg Ssb1 a day for 3 days ($n = 5$, Mean ± S.D.). No obvious organ damage or lesion was observed from Ssb1 nanoparticle treated mice (scale bar, 100 μm).

the source. Ssb1 and the deglycosylated metabolite has been reported to enhance liver targeting through inhibiting CYP3A4[57]. Besides, Saikosaponins, including Ssb1, could protect liver from $CCl_4$ induced liver injury[58]. As expected, Ssb1-alone treatment could inhibit α-SMA expression in in vitro experiments but was minimally effective in reversing the liver fibrosis of Balb/c mice and MnO$_2$@PLGA/Ssb1 exhibited more efficient therapeutic effect in in vitro and in vivo experiments.

In summary, we demonstrated that the hypoxia and OS of liver was an important factor in liver fibrogenesis, and hypoxia-induced TGF-β1 regulated CAT expression in HSCs. The constructed MnO$_2$@PLGA/Ssb1 nanodrug displayed capability to relieve liver hypoxia and OS and enhance the antifibrotic efficiency of Ssb1; thus, it can be a therapeutic approach to liver fibrosis treating. Based on this strategy, more promising solutions against hypoxia and OS may be applied in other fibrotic diseases, such as lung fibrosis and renal fibrosis.

## Methods

**Ethics statement**. All experiments of animals have received the approval of Ethics Committee of Zhejiang Chinese Medical University (Hangzhou, China, Project no. SYXK (浙) 2021-0012) and all animals have received good care. Paraffin-embedded human liver tissues were obtained from The First Affiliated Hospital, Guizhou University of Traditional Chinese Medicine, and were approved by Ethics Committee of the First Affiliated Hospital of Guizhou University of Traditional Chinese Medicine (Project no. K2021-066).

**Materials**. Ssb1 (Desit, Chengdu, China), PLGA (lactide/glycolide = 50/50; MW: 94 000 Da; MedChemExpress, USA), polyvinyl alcohol (PVA; BBI Co., Ltd., Shanghai, China), 2-(N-morpholine)ethanesulfonic acid (MES; Heowns Biochem LLC, Tianjian, China), KMnO$_4$ (Sinopharm Chemical Reagent Co., Ltd., Shanghai,

China), radio-immunoprecipitation assay (RIPA) buffer (Solarbio, Beijing, China), phenylmethanesulfonyl fluoride (PMSF, 1:100, Cell Signaling Technology, MA, USA), and 3-(4,5-dimethylthiazol-2-yl)-2,5-diphenyltetrazolium bromide (MTT; Solarbio, Beijing, China) were purchased and used as received. TUNEL, DCFH-DA and DiR iodide were purchased from Beyotime Ltd. (Shanghai, China). Fetal bovine serum (FBS), trypsin-EDTA solution (0.25 %), and Dulbecco's modified Eagle medium (DMEM) were obtained from Gibco (Burlington, Canada). All other chemicals and reagents of the highest purity available were obtained from commercial sources.

**Immunofluorescence analysis**. Immunofluorescent staining was performed on frozen sections (4 μm thick) fixed with the mixture of methanol and acetone. The sections were blocked with 5% goat serum, and incubated with special primary antibodies (Collagen I, 1:1000, 14695-1-AP, Proteintech; α-SMA, 1:1000, cs19245, Cell Signaling Technology (CST); HIF-1α, 1:1000, cs36169, CST; and CAT, 1:2000, 66765-1-Ig, Proteintech) at 4 °C overnight, followed with Alexa Fluor 488-labeled goat anti-rabbit IgG (H + L) secondary antibodies (1:200, A0423, Beyotime). Finally, slides were mounted with DAPI. All sections were observed and analyzed with a fluorescence microscope (ZEISS, AXIO SCOPE.A1). Paraffin-embedded human liver tissues were rehydrated and then stained using a Sirius Red staining Kit (RS1220, G-CLONE) to monitor collagen distribution.

**Cell culture**. Human HSC line LX-2 was purchased from Procell Life Science & Technology Co., Ltd. (Shanghai, China). Rat liver stellate cell line HSC-T6 was provided generously by Prof. Su Tao (Guangzhou University of Chinese Medicine). LX-2 and HSC-T6 cells were cultured in DMEM supplemented with 10% (v/v) FBS, and 1% penicillin-streptomycin in a humidified atmosphere containing 5% CO$_2$ at 37 °C.

**Hypoxia treatment**. For hypoxia stimulation, the HSCs were incubated with serum-free medium for 24 h. the culture plates were then incubated in a hypoxia incubator chamber (Billups-Rothenberg, CA, USA) flushed with a gas mixture containing 5% CO$_2$ and 95% N$_2$ until the oxygen tension dropped to 5%. The hypoxia incubator chamber was sealed and incubated at 37 °C for further 24 h or

48 h. In the control group, cells were cultured under normoxia (21% oxygen tension), and cells challenged by TGF-β1 (10 ng/mL) were cultured in normoxia as a positive control.

**Pharmaceutical inhibition**. Smad3 activation was inhibited using the specific inhibitor SIS3 (MACKLIN, s872443). For the inhibition experiments, the cells were incubated with SIS3 at 3 μM for 1 h before TGF-β1 (10 ng/ml, 24 h) incubation. After 24 h, cells were collected for WB and PCR text.

**RNA isolation and quantitative real-time PCR**. Total cellular RNA was extracted using Eastep Super Total RNA Extraction Kit, according to the instructions of the manufacturer (Promega, China). The quantification and concentration of RNA were measured using NanoDrop™ 2000 Spectrophotometer (Thermo Fisher Scientific, USA). cDNA was synthesized from 1 μg of total RNA using the Yfx Script First Strand cDNA Synthesis Kit (YIFEIXUE BIO TECH, China). Real-time PCR was performed using 2x SYBR Green Fast qPCR Master Mix and Light Cycle 96 System (Roche), according to the manufacturer's protocols. The primer sequences were shown in follows: Rat catalase I forward primer: 5′-CCCAGAAGCCTAA-GAATGCAA-3′; reverse primer: 5′-TCCCTTGGCAGCTATGTGAGA-3′; Rat b-actin forward primer: 5′-CCCGCGAGTACAACCTTCTTG-3′; reverse primer: 5′-TCATCCATGGCGAACTGGTGG-3′; Human catalase I forward primer: 5′-CTTCGACCCAAGCAACATGC-3′; and reverse primer: 5′-ATTTGGAGCAC-CACCCTGATT-3′; The relative expression level was calculated using the 2-ΔΔCt equation. All tests were performed at three biological repeats.

**Nrf2 knockdown**. Nrf2 shRNA plasmids were purchased from Shanghai Genechem Co. Ltd. (Shanghai, China), and sequence results are available in Supplementary Data 1. Nrf2 was knocked down by transfection with Nrf2 shRNA or an empty plasmid as control. Briefly, HSC-T6 cells were grown on 12 well plate to 60–70% confluence. HSC-T6 cells were incubated with the plasmids (1 μg) enfolded with the Lipofectamine 3000 Transfection Kit (Invitrogen, USA).

**Evaluation of intracellular ROS**. The intracellular level of ROS was monitored by using the fluorescent probe of DCFH-DA[59]. In a typical procedure, HSCs at a density of $5 \times 10^5$ cells/well in a glass-bottom dish were exposed to normoxia and hypoxia (5% O$_2$) with different treatments. About 24 h later, cells were incubated with 10 μM DCFH-DA and 100 nM Mito Tracker Deep Red FM in a working solution at 37 °C for 30 min in darkness. Then, the medium was removed and HSC cells were washed with PBS for three times. Fluorescent signals were analyzed with a CLSM system (Zeiss, LSM880, Germany) and CytoFlex flow cytometry (Beckmancoulter, CytoFlex, USA).

**Synthesis of MnO$_2$@PLGA/Ssb1 NPs**. Ssb1-loaded PLGA NPs (PLGA/Ssb1 NPs) were prepared through an emulsion solvent evaporation process[43]. PLGA and Ssb1 were dissolved in acetone at a concentration of 10 and 2 mg/mL, respectively. Then, 1 mL of the prepared acetone solution was added dropwise to 4 mL of Polyvinyl alcohol (PVA) solution (10 mg/mL) in an oil bath drop with a magnetic stirrer at a rate of 200 cycles per minute and 37 °C. The resulting solution was then stirred for 6 h to allow the acetone solution to evaporate completely. Finally, the NPs were centrifuged at 15,000×g for 10 min to remove the excess PVA and washed twice with ultrapure water to obtain the purified Ssb1/PLGA core.

To prepare MnO$_2$@PLGA/Ssb1 NPs, layered manganese oxide was formed on the surface of PLGA/Ssb1 NPs in MES buffer (pH 5.9) by oxidation-reduction method after adding 5 mM potassium permanganate for 30 min under ultrasonic conditions. Then, the NPs were centrifuged at 15,000 g for 10 minutes to remove possible free manganese oxide NPs and the MES, and washed twice with ultrapure water to obtain the purified MnO$_2$@PLGA/Ssb1 NPs.

**Characterizations of MnO$_2$@PLGA/Ssb1 NPs**. The hydrodynamic size distribution and surface potential of NPs were measured at 25 °C by a DLS (ZEN 3600 Zetasizer, Malvern). The surface morphology of NPs was observed under SEM and transmission electron microscopy. Elemental mappings were conducted using a transmission electron microscope (FEI Talos F200S). Manganese content was measured on an ICP-MS system (Thermo Fisher ICAP QC).

**Determination of Ssb1 loading**. The MnO$_2$@PLGA/Ssb1 NPs were first dissolved in acetonitrile, and the sonicated to ensure complete Ssb1 dissolution. Ssb1 concentration was determined by HPLC (Agilent Technologies, USA) under the following conditions. The HPLC column was an XDB-C18 column (4.6 × 250 mm 5.0 μm, Agilent Technologies) and the mobile phase comprised of acetonitrile and water. The column temperature was 30 °C and detected at 254 nm. The flow rate was 1.0 mL/min, and the injection volume was 10 μL. The drug-encapsulation efficiency was estimated based on the encapsulation efficiency (EE) relative to the total drug contained in the prescription.

$$EE\,(100\,\%) = (\text{Weight of the drug in the NPs})/(\text{Weight of drug}) \times 100\,\%$$

**In vitro drug release of MnO$_2$@PLGA/Ssb1 NPs**. For in vitro release study, PLGA/Ssb1@MnO$_2$ and PLGA/Ssb1 samples dissolved in ultrapure water were placed in dialysis membrane bags (MWCO: 1000; Scientific Research Special) and suspended in 2 mL of the release medium PBS (0.01 M) at pH 7.4. The release assay was performed at 37.0 ± 0.5 °C at a stirring rate of 150 rpm/min. After drawing 1 mL of the dissolution medium was drawn from the solution outside the dialysis bags at predetermined time intervals, the same volume of counterpart release medium solution was replenished. The concentration of Ssb1 released from NPs into the release medium was quantified by using an UV-vis assay.

**In vitro detection of O$_2$ generation**. Generated O$_2$ was measured with a JPSJ-605F dissolved-oxygen meter under an environment sealed with 3 mL of liquid paraffin to isolate the air. Before examination, the dissolved O$_2$ was ejected through bubbling nitrogen until the value decreased to 0.5 mg/L. Solutions (10 mL) containing H$_2$O$_2$ at different concentrations (0, 11, 33, and 44 mM) were added to MnO$_2$@PLGA/Ssb1 (Mn concentration: 233 μM) to generate O$_2$. Readings of the dissolved-oxygen meter were recorded every 10 s 50 times. Different concentrations of MnO$_2$@PLGA/Ssb1 (Mn concentrations: 0, 58, 176, and 233 μM) were detected with 44 mM H$_2$O$_2$, and the dissolved O$_2$ was recorded in the same way as above. After that, MnO$_2$@PLGA/Ssb1 (Mn: 117 μM) catalyzation of H$_2$O$_2$ (22 mM) in pH 5.0, pH 6.5, and pH 7.2 buffers was detected through the dissolved O$_2$.

**Antifibrotic effect of Ssb1**. MTT was used to determine the cytotoxicity of Ssb1 on the quiescent and TGF-β1-activated HSC-T6/LX-2 cells. HSC-T6 and LX-2 cells were seeded onto 96-well culture plates at a density of 1×10$^4$ cells/well separately and incubated at 37 °C. After 12 h, cells were transferred to serum-free DMEM for quiescent HSCs and were treated with 10 ng/mL TGF-β1 for 30 min for activated HSCs. Cells were then treated with different concentrations of Ssb1 and co-cultured for 24 h. Then, 20 μL of MTT (5 mg/mL) was added and incubated for 4 h. When formazan formed, the culture was removed and 150 μL DMSO was added to dissolve the formazan. Optical-density values were measured at 570 nm by using a microplate reader (Bio-Rad).

HSC-T6 and LX-2 cells were seeded onto six-well culture plates at a density of $5 \times 10^5$ cells/well and incubated at 37 °C. After 12 h, cells were transferred to serum-free DMEM and treated with 10 ng/mL of TGF-β1 for 30 min, and then added with 5, 10, 15 μM Ssb1 for 24 h. Expression of Collagen I, α-SMA and Caspase 3 were quantified using WB.

**In vitro antifibrotic effect**. To test the antifibrotic efficiency of the various Ssb1-containing formulations, HSC-T6 and LX-2 cells were seeded onto six-well culture plates at a density of $5 \times 10^5$ cells/well and incubated at 37 °C. After 12 h, cells were transferred to serum-free DMEM and treated with 10 ng/mL of TGF-β1 for 30 min, and then added with different formulations (15 μM Ssb1) for 24 h. Before the drug intervention, cells were synchronized in serum-free DMEM for 24 h. For HSC-T6 and LX-2 cells, the medium was then replaced by different solutions: (1) serum-free DMEM, (2) serum-free DMEM with 10 ng/mL TGF-β1, (3) serum-free DMEM with 10 ng/mL TGF-β1 and 15 μM Ssb1, (4) serum-free DMEM with 10 ng/mL TGF-β1 and MnO$_2$ NPs (loaded with 9.9 μM Mn); (5) serum-free DMEM with 10 ng/mL TGF-β1 and PLGA/Ssb1 (loaded with 15 μM Ssb1); (6) serum-free DMEM with 10 ng/mL TGF-β1 and MnO$_2$@PLGA/Ssb1 (loaded with 15 μM Ssb1, 9.9 μM Mn). After treatment for 24 h, the cells were used to evaluate the expression of fibrotic-related proteins using WB and immunofluorescent staining analyses. For further experiments, cells were subjected to 24 h of hypoxia or 50 μM H$_2$O$_2$ treatments, whereas other sections were left untreated.

**In vivo fluorescence imaging**. The tissue distribution of NPs was determined using DiR-loaded NPs instead of Ssb1-loaded NPs[50,60]. Mice were anesthetized and free DiR, PLGA/DiR, and MnO$_2$@PLGA/DiR, (DiR/body weight = 2.5 mg/kg, 100 μL) were intravenously injected into mice. At 0.5, 1, 4, and 8 h after injection, mice were anesthetized using isoflurane and subjected to in vivo fluorescence imaging with an in vivo imaging system (AniView600) at excitation and emission wavelengths of 745 and 800 nm, respectively. After 8 h, mice were sacrificed and then their heart, livers, lungs, spleens, and kidneys were removed for ex vivo imaging using the same system.

**In vivo biodistribution of Ssb1 and MnO$_2$**. For biodistribution measurement, healthy Balb/c mice were i.v. injected with Ssb1, MnO$_2$, and PLGA/Ssb1 (100 μL per mouse; dosage = 8.25 mg/kg in terms of Ssb1 weight concentrations). Control groups were i.v injected with 100 μL of PBS. Six mice were used per group. Their major organs including heart (H), liver (L), spleen (S), lung (Lu), and kidney (K), were collected at 8 h post-injection. The organs were weighed. Tissues were homogenized in PBS. Ssb1 was extracted from tissues by protein precipitation. About 200 μL of homogenate was treated with 400 μL of acetonitrile (4 °C) and vortex mixed for 3 min. The mixtures were centrifuged at 15,000×g for 5 min (4 °C). The supernatants were collected and analyzed.

The distribution of Ssb1 in various formulations was analyzed by LC-MS/MS. Negative mode was operated on Waters Xevo G2-XS QTOF mass spectrometer (Waters, Manchester, UK) and was coupled with an Acquity I-class UPLC system (Waters, Milford, USA, Acquity HSS T3 column). The mass analyzer scanned over

a mass range of 50–2000 Da in full scan with a scan time of 0.1 s for fast DDA, and over m/z 150–1300 for MS/MS by the same scan time. LockSpray was conducted with 10 ng/L leucine-enkephalin solution at a flow rate of 10 μL/min. The UPLC elution gradient for separation was as follows: 20% of solvent B (acetonitrile) at 0 min; 52% of solvent B at 12 min; 90% of solvent B at 22 min; and 20% of solvent B at 30 min. The flow rate was 0.3 mL/min. Data analysis was performed using MassLynx V4.1 software (Waters, Milford, USA). Certificate of analysis was performed on ACQUITY UPLC® HSS T3, 1.8 uL (Waters, Manchester, UK).

The contents of Mn were measured by ICP-MS. Organs (50 μL) were transferred to scintillation vials. Nitric acid (100 μL) was added, the vials were sealed, and the organs were left to digest for 20 h at 37 °C. After diluting 100 μL of organ digest with 1 mL of ultrapure water Mn content was calculated using ICP on the injection volumes used for each animal (0.16–0.22 μmol per mouse; on average, the administered $MnO_2$ dose was 10 μmol/kg with 0.2 μmol/mouse)

**Animal experiments**. All experimental procedures received the approval of the institutional and local committee on the care and use of animals in Zhejiang Chinese Medical University (Hangzhou, China). Whole animals were given humane care according to the National Institutes of Health guidelines. Male Balb/c mice weighing about 18–22 g were purchased from the Center of Experimental Animal of Zhejiang Chinese Medical University (Hangzhou, China). Mice were fed at 25 °C and placed in 40–60% humidity with a 12 h light/dark cycle and free access to water and laboratory chow unless otherwise specified. All mice were adapted to their environment for 1 week before starting experiments. A mixture of $CCl_4$ (0.06 mL/20 g body weight) in olive oil (2:3 (v/v)) was utilized to trigger hepatic fibrosis in mice by subcutaneous injection twice weekly for 5 weeks (W) or 8 W. Mice in untreated groups were administrated with the same volume of corn oil as healthy control.

For therapeutic study, after 5 W of 40% $CCl_4$ administration, male Balb/c mice were randomly divided into seven groups (5 mice per group). Mice were then treated with the following solutions once a week for 3 W: group 1, control group; group 2, PBS through tail-vein injection; groups 3–6, fibrotic mice (Ssb1/body weight = 10:1 mg/kg) intravenously injected with free Ssb1, $MnO_2$, PLGA/Ssb1, and $MnO_2$@PLGA/Ssb1 suspended in 100 mL of sterile PBS, respectively. Mice in groups 2–6 were subcutaneously injected with $CCl_4$ once a week for 6–8 W, and control group was administrated with same volume of olive oil.

One day after the last injection, all mice were executed and their blood and liver were collected. Serum levels of alkaline phosphatase (ALP), aspartate aminotransferase (AST), and alanine aminotransferase (ALT) were measured with an automatic biochemical analyzer. Four indicators (HA, LN, PIIINP, and IV-C) of serum liver fibrosis were analyzed with an ELISA kit. Liver tissues were dissected and stored in 10% formalin for subsequent histological analysis. Part of the liver was stored in liquid nitrogen for subsequent Western blot analysis.

**Measurement of liver $H_2O_2$ Level**. $H_2O_2$ accumulation was determined using a $H_2O_2$ Assay Kit (Nanjing Jiancheng Bioengineering Institute, China). Snap-frozen liver tissue was homogenized with saline. $H_2O_2$ concentration was then measured following the manufacturer's instructions.

**In vitro and in vivo biosafety assays**. The in vitro cytotoxicity evaluation of $MnO_2$@PLGA/Ssb1, PLGA/Ssb1, and Ssb1 was conducted by MTT assay. HSC-T6, and LX-2 cells were seeded onto 96-well plates at a density of $1 \times 10^4$ cells per well and grown in 100 μL of medium supplemented with 10% (v/v) FBS. After 24 h at 37 °C, the medium was removed and the cells were incubated with 200 μL of fresh medium containing various concentrations of PLGA/Ssb1@MnO₂, PLGA/Ssb1, or free Ssb1 solution for 24 h. Then, 20 μL of MTT solution (5 mg/mL) was added to each well, followed by another 4 h incubation at 37 °C. Finally, the medium containing MTT was removed and 150 μL of DMSO was added to dissolve the formazan crystals. $OD_{570\ nm}$ was measured with a microplate reader (Synergy H1, Biotek, USA). All experiments were conducted in quadruplicate. Untreated cells served as a control.

Acute toxicity was evaluated to ensure the biosafety of the nanosystem. Balb/c mice were respectively divided into five groups (n = 5): PBS (Untreated), Ssb1, $MnO_2$, PLGA/Ssb1, and $MnO_2$@PLGA/Ssb1. Mice in all groups were locally intravenously injected with different solutions (0.1 mL; including an equivalent Ssb1 concentration of 1.65 mg/mL) once a day for three consecutive days. Twenty-four hours after the last injection, mice were sacrificed, and the main organs (heart, liver, spleen, lung, and kidney) were dissected and stored in 10% paraformaldehyde for subsequent for H&E staining. Blood was collected and analyzed for AST, ALT, and ALP to assess in vivo acute toxicity.

**Western blot analysis**. Total proteins were extracted from cells by using radio-immunoprecipitation assay (RIPA) buffer (R0010, Solarbio) supplemented with Phenylmethanesulfonyl Fluoride (PMSF, 8553, CST). Protein concentration was measured with a BCA Protein Assay Kit (P0011, Beyotime). Protein was loaded on an SDS-PAGE gel (sodium dodecyl-polyacrylamide gel electrophoresis) and transferred to polyvinylidene difluoride (PVDF) membranes (IPVH00010 Merck Millipore, Billerica, MA, USA) by electroblotting. Membranes were blocked by

incubating for 1 h at room temperature (RT) in PBS-buffered saline 0.1% Tween-20 containing 5% skim milk. They were then incubated first with the appropriate primary antibodies: a-SMA (1:1000, 14395-1-AP, Proteintech) CAT (1:2000, 21260-1-AP, Proteintech), HIF-1α (1:1000, AF1009, Affinity), Caspase-3 p12 (1:1000, ab179517, Abcam), TGF-β1 (1:1000, ab215715, Abcam), Collagen I (1:1000, 14695-1-AP, Proteintech), Collagen I (1:1000, ab34710, Abcam) GAPDH (1:5000, AF1186, Beyotime), α-Tubulin (1:5000, 11224-1-AP, Proteintech), Foxo3a (1:1000, AF7624, Affinity), CAT (1:2000, 66765-1-Ig), Nrf2 (1:1000, 16396-1-AP), and β-actin (1:5000, 66009-1-Ig) in primary antibody diluent (P0023A, Beyotime) at 4 °C overnight. Afterward, the membranes were incubated with HRP-conjugated anti-rabbit IgG (1:5000, ab97051, Abcam) or anti-mouse IgG (1:5000, BA1050, Boster) secondary antibodies for 1 h at room temperature. Immunoreactive bands were visualized using ChemiDoc™ Touch image system (ChemiDoc™ Touch, Bio-Rad, CA, USA). In some experiments, the previous primary antibody and secondary antibody were stripped from the PVDF membrane with stripping buffer (SW3022, Solarbio) for 30 min. After being re-blocked, the membrane was re-incubated with another primary antibody at 4 °C overnight. The following steps were the same as described above. Densitometrically quantified using ImageJ software and protein-expression levels were normalized against GAPDH.

**Histological analysis**. Liver tissues were soaked in 10% formalin and embedded in paraffin. The obtained tissues were pretreated, dehydrated, and embedded in paraffin. Paraffin-embedded liver tissues were cut into 4 μm sections. After deparaffinization and hydration, sections were stained. H&E was used for pathological assessments according to the organizational structure. Masson staining was used to evaluate collagens. A microscope (ZEISS Axio Vert. A1) was used to take photographs of these stained sections at random fields.

**Statistics and reproducibility**. All tests were repeated more than three times to ensure that the all results were accurate. The quantitative data were presented as mean ± standard deviation (S.D.). Data are presented as mean and analyzed for comparison of two groups by Student's t test or Mann–Whitney U test, respectively. All graphs were drawn using GraphPad prism 8 version software. $P < 0.05$ was recognized to be statistically significant.

**Reporting summary**. Further information on research design is available in the Nature Portfolio Reporting Summary linked to this article.

## Data availability

All data supporting the findings of this study are available within the article and its supplementary information files (Supplementary information and Supplementary Data 1–3) or from the corresponding author upon reasonable request. The transcriptome sequence data have been submitted to the Sequence Read Archive (SRA) databases under BioProject number PRJNA916271.

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

## Acknowledgements

We thank Dr. Tao Su for supporting rat liver stellate cell line HSC-T6. This work was financially supported by the National Natural Science Foundation of China (Nos. 52003246, 81922073 and 81973481); the Traditional Chinese Medicine Key Scientific Research Fund Project of Zhejiang Province (Nos. 2018ZY004, 2022ZQ032 and 2021ZZ009); and the Youth Natural Science Program of Zhejiang Chinese Medical University (2021JKZKTS007A).

## Author contributions

M.P. and G.C. conceived the study and all authors were involved in the experiment design. Q.L., M.H., and K.W. were responsible for the HPLC and LC-MS experiments. M.S., H.D., and X.H. managed in vitro experiments. Q.Y. assisted animal model construction and intravenous injection. H.K. provided professional assistance. D.T. and Z.S.

provided clinical samples. M.P. and M.S. wrote this manuscript and G.C. revised the manuscript.

## Competing interests

The authors declare no competing interests.
