## [Peer Review File · Communications Biology]

Reviewers' comments:

Reviewer #1 (Remarks to the Author):

The paper by Peng et al. designed a multifunctional nanosystem combining CAT-like MnO₂ and anti-fibrosis Saikosaponin b1 (Ssb1) for antifibrotic therapy. The authors have done a lot of experiments to prove the feasibility of their idea. In my opinion, the paper can be published in Communications Biology after modification. I only have a few small concerns.

The authors should provide more research background of Nano materials in the application of liver fibrosis, especially the application of mimics of catalase in this field.

The catalytic activity of Nanomaterials should be quantitatively characterized, such as TON value.

TEM should preferably be a picture with a small magnification.

The images of Western should be quantified.

Many pictures in the article are crowded.

Reviewer #2 (Remarks to the Author):

This study examines the potential synergistic effect of combining MnO₂ with Ssb1 within a PLGA nanoformulation to achieve enhanced anti-fibrotic effect on liver. The authors first established the importance of catalase as a regulatory control for oxidative stress. By suppressing hydrogen peroxide production, fibrogenic activity was diminished. Henceforth, the authors evaluated that MnO₂ and Ssb1 were significant in arresting hydrogen peroxide release, even under the pressure of hypoxia as well as TGF- β 1 induction. The nanoformulation comprising of both MnO₂ and Ssb1 encapsulation replicated the observation using both in vitro and in vivo models. While efficacy was undisputable, the hypothesized synergism remains unclear as the superiority of the nanoformulation vs. MnO₂ and MnO₂-NP is not conclusive from the results obtained. Specific comments are as shown below:

1. First part of the work was to establish the centrality of catalase levels in the manifestation of liver fibrosis. While adding TG- β 1 showed a direct suppression of catalase, it will be more convincing if there was a reciprocal experiment: A loss-of-function experiment will help to ascertain that TGF- β 1 is not just associated with catalase, but is responsible for catalase regulation.
2. Nrf2 is another key regulator of catalase. In the transcriptomics analysis, it may also be useful to highlight if Nrf2 perturbation could contribute to the changes in catalase expression and activities, as another effector arm that respond to both hypoxia and TGF- β 1 induction.
3. In the discussion, more could be said about the known mechanism of Ssb1, to substantiate the conclusions drawn in this study.
4. In Figure 3, much of the anti-oxidants effect of the formulation occurred within the timescale of 15 min. How would this affect the in vivo activities, where injection is intravenous, and it may take a longer duration before the formulation can accumulate sufficiently in the target organs?
5. In Figure 4, the results show pronounced efficacy with MnO₂ alone, whereas PLGA/Ssb1 and MnO₂@PLGA/Ssb1 may not necessary demonstrate any enhancement. Therefore, synergistic effect may require further qualification and validation.

Response to referees:

Reviewer 1:

The paper by Peng et al. designed a multifunctional nanosystem combining CAT-like MnO₂ and anti-fibrosis Saikosaponin b1 (Ssb1) for antifibrotic therapy. The authors have done a lot of experiments to prove the feasibility of their idea. In my opinion, the paper can be published in Communications Biology after modification. I only have a few small concerns.

1. The authors should provide more research background of Nano materials in the application of liver fibrosis, especially the application of mimics of catalase in this field.

Reply: Background of CAT-like nanomaterials was added in Page 4, line 2 as suggested.

2. The catalytic activity of Nanomaterials should be quantitatively characterized, such as TON value.

Reply: The catalytic activity of nanomaterials in different conditions was quantified by O₂ generation in **Fig. 3h-j**.

3. TEM should preferably be a picture with a small magnification.

Reply: Revised as suggested (**Fig. 3b**).

4. The images of Western should be quantified.

Reply: WB was quantified as suggested and displayed in **Supplementary Fig. 1, 2, 4b-e, 5b, 6b-c**.

5. Many pictures in the article are crowded.

Reply: Crowded pictures (**Fig. 2, Fig. 4, Fig. 6**) have been revised as suggested.

Reviewer 2:

This study examines the potential synergistic effect of combining MnO₂ with Ssb1 within a PLGA nanoformulation to achieve enhanced anti-fibrotic effect on liver. The authors first established the importance of catalase as a regulatory control for oxidative

stress. By suppressing hydrogen peroxide production, fibrogenic activity was diminished. Henceforth, the authors evaluated that MnO₂ and Ssb1 were significant in arresting hydrogen peroxide release, even under the pressure of hypoxia as well as TGF- β 1 induction. The nanoformulation comprising of both MnO₂ and Ssb1 encapsulation replicated the observation using both in vitro and in vivo models. While efficacy was undisputable, the hypothesized synergism remains unclear as the superiority of the nanoformulation vs. MnO₂ and MnO₂-NP is not conclusive from the results obtained. Specific comments are as shown below:

1. First part of the work was to establish the centrality of catalase levels in the manifestation of liver fibrosis. While adding TG- β 1 showed a direct suppression of catalase, it will be more convincing if there was a reciprocal experiment: A loss-of-function experiment will help to ascertain that TGF- β 1 is not just associated with catalase, but is responsible for catalase regulation.

Reply: Many thanks for the suggestion. TGF- β 1/Smad3 pathway has been reported to decrease expression of FoxO3a (forkhead box type O3a) in cardiac fibrosis, which positively regulated CAT mRNA through binding to its promoter (*BBA-Molc. Cell Res.* 1867, **2020**, 118695; *J. Biol. Chem.* 283, **2008**, 29730-29739). The Smad3-specific inhibitor SIS3 was used to block TGF- β 1/Smad3 pathway, and significant recovery of Foxo3a and CAT expression was observed in SIS3 treated active HSCs, indicating that TGF- β 1 could down regulate CAT through Foxo3a (Page 6, line 15, line 22; **Fig. 2e** and **2f**).

2. Nrf2 is another key regulator of catalase. In the transcriptomics analysis, it may also be useful to highlight if Nrf2 perturbation could contribute to the changes in catalase expression and activities, as another effector arm that respond to both hypoxia and TGF- β 1 induction.

Reply: Nrf2 was silenced through Nrf2 shRNA as suggested. Hypoxia and TGF- β 1 incubated HSCs showed decreased Nrf2 expression. After Nrf2 was silenced, CAT was down regulated as a result, indicating that Nrf2 contributed to CAT regulation that respond to both hypoxia and TGF- β 1 induction (Page 7, line 4; **Supplementary Fig. 2**).

3. In the discussion, more could be said about the known mechanism of Ssb1, to substantiate the conclusions drawn in this study.

Reply: Background of Ssb1 mechanism was added in page 15, line 16 as suggested.

4. In Figure 3, much of the anti-oxidants effect of the formulation occurred within the timescale of 15 min. How would this affect the in vivo activities, where injection is intravenous, and it may take a longer duration before the formulation can accumulate sufficiently in the target organs?

Reply: Thanks for this suggestion. During in vivo circulation, NPs were supposed to slowly release Ssb1 (in vitro constantly release within 28 h), and nearly no H₂O₂-splitting reaction was occurred due to low H₂O₂ content of blood. The H₂O₂-splitting capability was only efficient in the presence of sufficient H₂O₂ (Fig. 3h). Therefore, the anti-oxidants effect of the nanoplatform was mainly displayed in the target organ.

5. In Figure 4, the results show pronounced efficacy with MnO₂ alone, whereas PLGA/Ssb1 and MnO₂@PLGA/Ssb1 may not necessary demonstrate any enhancement. Therefore, synergistic effect may require further qualification and validation.

Reply: The synergistic effect of NPs has been further determined in animal experiments and displayed enhanced antifibrotic effect. In Fig. 7c and 7g, HIF-1 α /Collagen I expression and serum ALT/AST activity of MnO₂@PLGA/Ssb1 group were significantly lower compared with MnO₂ alone, indicating that MnO₂@PLGA/Ssb1 showed synergistic effect against CCl₄-induced liver fibrosis.

REVIEWERS' COMMENTS:

Reviewer #1 (Remarks to the Author):

The authors have resolved all my concerns and I recommend the publication of the paper in its present form.

Reviewer #2 (Remarks to the Author):

The authors have thoroughly addressed the queries raised. In particular, they should be applauded for the efforts to add in new experimental results on Nrf2 shRNA to demonstrates the concurrent influence of both Nrf2 and TGFb in the regulation. They haev already added more details to show how the nanoconstructs present enhanced efficacy to simply the pharmacological action of MnO2.

I have no further concern.